# Cardiopulmonary Exercise Test Parameters in Athletic Population: A Review

**DOI:** 10.3390/jcm10215073

**Published:** 2021-10-29

**Authors:** Reza Mazaheri, Christian Schmied, David Niederseer, Marco Guazzi

**Affiliations:** 1Department of Sports and Exercise Medicine, Division of Cardiology, Tehran University of Medical Sciences, Tehran 1419733141, Iran; mazaheri_md@tums.ac.ir; 2Department of Cardiology, University Heart Center, University Hospital Zurich, University of Zurich, 8091 Zurich, Switzerland; christian.schmied@usz.ch (C.S.); david.niederseer@usz.ch (D.N.); 3Department of Health Sciences, San Paolo University Hospital, University of Milano, 20142 Milan, Italy

**Keywords:** athletes, cardiopulmonary exercise test, exercise physiology, sports performance

## Abstract

Although still underutilized, cardiopulmonary exercise testing (CPET) allows the most accurate and reproducible measurement of cardiorespiratory fitness and performance in athletes. It provides functional physiologic indices which are key variables in the assessment of athletes in different disciplines. CPET is valuable in clinical and physiological investigation of individuals with loss of performance or minor symptoms that might indicate subclinical cardiovascular, pulmonary or musculoskeletal disorders. Highly trained athletes have improved CPET values, so having just normal values may hide a medical disorder. In the present review, applications of CPET in athletes with special attention on physiological parameters such as VO_2_max, ventilatory thresholds, oxygen pulse, and ventilatory equivalent for oxygen and exercise economy in the assessment of athletic performance are discussed. The role of CPET in the evaluation of possible latent diseases and overtraining syndrome, as well as CPET-based exercise prescription, are outlined.

## 1. Introduction

Cardiopulmonary exercise testing (CPET) provides a full assessment of the physiologic responses of the pulmonary, cardiovascular, muscular, and cellular oxidative systems to exercise [1,2]. Sports physicians and physiologists try to identify the effects of exercise on athletes’ organs to figure out their conditioning level and plan training programs to develop elite athletes for the team. Although CPET has become a generally well-accepted method to assess organ system adaptations to chronic regular exercise, its useful applications for more comprehensive assessments in the athletic population are not universally widespread yet.

The proposed indications for CPET in apparently healthy athletes are [3,4,5,6]:-Measurement of baseline fitness and assessment of physiological function of body’s systems;-Evaluation of the integrated cardiopulmonary response in asymptomatic athletes with cardiac diseases;-Diagnosis of latent disease and/or evaluation of minor nonspecific symptoms;-Exercise prescription with specific purposes in different sports disciplines.

Athletes mainly recognize CPET assessment just for the measurement of their cardiorespiratory fitness (CRF) that mainly provides aerobic endurance performance expressed as maximal oxygen consumption (VO_2_max). VO_2_max measured during a graded maximal exercise is the most significant parameter to assess the cardiopulmonary capacity, especially in endurance sports [7], but it should be validated by completing a short constant work rate phase at higher intensities than the VO_2_max work rate achieved during the ramp tests [8]. There are additional parameters that extend the number of opportunities to comprehensively study the integrated organ system response to maximal effort. The most remarkable is likely the ventilatory threshold (VT) that indicates aerobic power and is related to the lactate threshold in endurance and even resistance exercises [9]. Moreover, critical power (CP) is a widely used parameter for training that represents threshold intensities associated with the upper limit for prolonged aerobic exercise and signifies high to severe exercise intensity domain [10].

The series of variables are however consistent, and a careful analysis of each one yields to the best information in the time point assessment of physical performance and response to training programs. Oxygen pulse, defined as the ratio between VO_2_ and heart rate (HR), indicates stroke volume and peripheral vascular perfusion/extraction response to exercise and reflects the maximal aerobic capacity. Ventilatory efficiency expressed by the relationship between minute ventilation (VE) to carbon dioxide (VCO_2_) production (VE/VCO_2_ slope) and/or partial pressure of end-tidal carbon dioxide (PETCO_2_) at rest and during exercise, represents a match of ventilation and perfusion within the pulmonary system [11]. These quite underused parameters require a specific evaluation and interpretation.

Regular exercise has favourable cardiovascular benefits, but competitive athletes usually perform intense training for prolonged periods which exposes their cardiovascular system to increased levels of strain [12]. Intense exercise may trigger adverse cardiac events in an asymptomatic athlete with latent cardiac disease. Moreover, athletes sometimes experience minor symptoms such as dizziness, palpitation, or chest tightness that might indicate some subclinical cardiovascular or pulmonary disorders. In these instances, CPET analysis may be of additional help in clarifying the underlying causes and abnormalities.

In this review, the applications of CPET in athletes with an emphasis on physiological parameters and their implication in the assessment of athletic performance are discussed. Secondly, the role of CPET in the evaluation of minor nonspecific symptoms to determine and/or distinguish cardiac and pulmonary conditions that might limit sports participation is discussed. Finally, recommendations on exercise prescription for athletes based on the CPET results are summarized at the end of the paper.

## 2. Cardiorespiratory Fitness

### 2.1. VO_2_max

VO_2_max is the maximum oxygen uptake of the human body that defines the maximal amount of energy accessible by aerobic metabolism at peak exercise [13]. It is a standard for quantifying CRF [14] and may reflect the limits of the cardiopulmonary system to maximal exercise. The term VO_2_max implies an individual’s physiological limit that is achieved and sustained for a specified period during maximal effort [15]. Athletes of different sport disciplines present with a wide range of VO_2_max, so for better inter-individual comparisons, it is better to express it as percent-predicted value or in millilitres of oxygen per kilogram of body weight per minute (mL/kg/min) [15,16]. Since the ideal body weight could be entirely different between disciplines, it makes sense to use fat-free mass instead of body weight for interdisciplinary comparison of athletes.

Expected values differ between male and female athletes at any given age and on different exercise test modalities. Accurate interpretation of VO_2_max should be made with the knowledge of what is expected for an individual athlete. To facilitate sports counselling, it is critical to have validated reference values in target population [14]. Athletes have higher amounts, usually more than 120% of the predicted VO_2_max of healthy untrained individuals, therefore, the interpretation of the results have to be done cautiously, as it might mask some latent disorders or potential physiological impairments. The time course of VO_2_ recovery after exercise is an essential parameter that must be considered in the athletic population [17]. In highly trained athletes, recovery of VO_2_ is more rapid, and just as depicted in Figure 1, the athlete with a higher exercise economy has a faster VO_2_ recovery rate than his counterpart with less efficient cardiovascular function.

There are considerable differences in the VO_2_max of individuals, and numerous genetic variants have been found to be associated with these variations. Studies have reported that genetic components and inheritance account for 44 to 72% of the baseline VO_2_max (mL/kg/min) in sedentary subjects [18]. Exercise training improves cardiorespiratory fitness by about 10–25% in previously sedentary individuals [15,19]. This improvement varies greatly between individuals even with a standard exercise training program. Genetic factors determine almost 50% of this VO_2_ response to training [19,20,21]. The presented evidence implies the importance of baseline VO_2_max measurements to find talented young athletes and to consult with athletes about their maximum achievable performance in different sport disciplines.

### 2.2. Ventilatory Threshold (VT)

During incremental exercise, there is a point at which muscles and blood lactate increase due to the rate of lactate production being higher than disposal [22]. The metabolic rate at which excess carbon dioxide (CO_2_) develops proportionally to the muscle and blood bicarbonate decreasing rate as a consequence of buffering metabolic acidosis is the ventilatory threshold [22]. This excess CO_2_ makes VE increase more steeply relative to the increase in VO_2_ [13,15]. Therefore, VT is a point, at which VCO_2_/VO_2_ slope becomes steeper; the ventilatory equivalent for oxygen (VE/VO_2_) begins to increase while the ventilatory equivalent for carbon dioxide (VE/VCO_2_) remains stable [23] (Figure 2).

VT is expressed as VO_2_ (mL/kg/min) or percentage of VO_2_max and compared with VO_2_max, is better correlated to athletic endurance performance [24,25]. It usually occurs at 45% to 65% of VO_2_max in healthy untrained subjects [15,23] and at a higher percentage (close to 90% of VO_2_max) in highly endurance-trained athletes [25]. It has been shown that after training, there is an increase in VO_2_ at VT by about 10–25% in sedentary individuals [15]. The weighted mean heritability of submaximal stamina and endurance test performance is 49% and 53% respectively [18]. This evidence shows the importance of genetic predisposition for VT and submaximal endurance performance and its potential application in identifying talents in sports.

It has recently been proposed that the lactate threshold (LT) could be used to set the training load in resistance exercises [9]. Resistance training promotes muscle hypertrophy, strength and power, and the intensity of exercise is the most important component in this way. Studies have identified the LT in strength training exercises at intensities ranging from 27% to 36% of a maximum repetition (1RM) [9]. Exercise at this level of intensity could be optimal training in sports requiring strength and power. To verify the association between LT and VT, investigators identified the VT with positive correlation, agreement and at the same intensity of exercise with LT during an incremental resistance exercise test [26,27]. Although there exists a slight difference between VT and arterial blood lactate accumulation with VT occurs earlier in dynamic exercise [22,28], yet VT can measure both the endurance and resistance performance in athletes.

The modality of exercise test (treadmill or cycle) and the population under investigation potentially influence the VT response to exercise [15]. In trained subjects, VT is significantly higher on the treadmill than cycle ergometer but not in untrained individuals [29]. This difference should be taken into consideration, especially when we want to make comparisons among athletes in different sports disciplines. In a study on 29 male competitive triathletes, Hue O et.al. showed that the VT values on treadmill running were lower than the values reported for elite distance runners [30].

## 3. Further Key CPET Parameters

### 3.1. Oxygen Pulse

The oxygen pulse (O_2_ pulse) is the ratio of VO_2_ in mL/min and HR in beats/min, expressed as mL/beat [23]. According to the Fick equation, VO_2_ = (HR × SV) × C(a-v)O_2_, where SV is stroke volume and C(a-v)O_2_ is the arterio-venous oxygen difference. Thus, the O_2_ pulse provides an estimate of stroke volume and peripheral vascular perfusion/extraction response to exercise [15]. Normal values at rest range from 4–6 mL/beat and increase up to 10–20 mL/beat at maximal exercise [23]. Athletes demonstrate a 10%–15% increase in ventricular cavity size and enhanced cardiac filling in diastole that augment their stroke volume (SV) compared with individuals of similar age and size [12,31]. They reveal increased mitochondrial oxidative capacity and capillarity within the skeletal muscle, which results in higher C(a-v)O_2_ during exercise [32]. As a result, O_2_ pulse is higher in trained athletes, but the reference values in this population are yet to be determined.

Central [SV] and peripheral [C(a-v)O_2_] adaptations to exercise result in higher O_2_ pulse in trained subjects. As mentioned earlier, in calculating the O_2_ pulse, VO_2_ should be in mL/min. So, for making reliable comparisons between athletes, the weight and height of the athletes have to be taken into account. Therefore, the authors of the present paper recommend calculating the O_2_ pulse in relation to body surface area (BSA). Since the relative contribution of SV to cardiac output is paramount during the early and intermediate phases of exercise [4,15], the amount of O_2_ pulse/BSA at submaximal levels (e.g., 50% or 75% of the VO_2_max) demonstrate more central (cardiac, i.e., SV) adaptations and the maximal value shows both central and peripheral (cardiovascular and muscular perfusion/extraction) adaptations to exercise.

Different patterns of adaptations according to O_2_ pulse response to incremental exercise are displayed in Table 1. This approach could be a precise manner to compare athletes with different conditioning levels. It guides sports physicians and athletic trainers to distinguish elite athletes and to prescribe the appropriate training program to focus on central (cardio-pulmonary) or peripheral (skeletal muscles) structures based on the standard requirements for any sports discipline.

### 3.2. Ventilatory Equivalents (VE/VO_2_ and VE/VCO_2_)

Athletic performance requires proper integration of cardiovascular, pulmonary, and skeletal muscle physiology. Physiological deficiency of any of these systems diminishes VO_2_ and increases ventilatory equivalents [15]. Enhanced mitochondrial function is a result of chronic exercise training [24], and this improvement is valuable for the effective production of adenosine triphosphate (ATP) through oxidative phosphorylation. Inadequate adaptation to exercise reduces the oxidative capacity of skeletal muscles and makes them rely on anaerobic glycolysis for ATP production, which leads to lactic acid accumulation early in exercise. Very deconditioned individuals and patients with mitochondrial myopathy might show the same manifestations but exaggeratedly [33].

Early lactic acidosis is reflected in CPET as a low VT, which is demonstrated by a rapid increase in VE/VO_2_ and respiratory exchange ratio (RER). Arterio-venous oxygen difference might also be lower in athletes with unfavourable mitochondrial adaptations resulting in a reduced peak O_2_ pulse. Some studies support the improvement of these parameters in fit individuals by demonstrating a decreased submaximal VE/VO_2_ and increased peak O_2_ pulse after exercise training [34,35]. Figure 3 depicts the ventilatory equivalents of two athletes with different mitochondrial adaptations.

The slope of the VE/VCO_2_ relationship and PETCO_2_ during an incremental exercise test represent the matching of ventilation and perfusion within the pulmonary system, and they are determinants of ventilatory efficiency in subjects. Studies have revealed a lack of relationship between ventilatory efficiency evaluated by VE/VCO_2_ slope and sports performance in athletes [36,37].

The relationship between VO_2_ and the log scale of VE represents the oxygen uptake efficiency slope (OUES) and expresses the ventilatory requirement for O_2_ [16,23]. Many investigators found it useful in the evaluation of fitness level, and reference values have been proposed [38], but a broadly accepted threshold to define normal response has not been clearly established. Training induced changes in OUES are variable and not sensitive enough to show the improvement of fitness after training [39].

### 3.3. Exercise Economy (ΔVO_2_/ΔWorkload)

Exercise economy is defined as the energy expenditure for given absolute exercise intensity and is expressed as VO_2_ at a given physical work or power output [4,24,40,41]. Remarkable economy means lower VO_2_ for given power output and is an advantage in endurance performance because it results in the utilisation of a lower percentage of VO_2_max for particular exercise intensity. Low VO_2_max scores can even be compensated by remarkable economy [24].

The importance of exercise economy has been described in different athletes such as runners and soccer players [41,42], and the improvements could be achieved by endurance training through improved muscle oxidative capacity and changes in motor unit recruitment patterns. Researches support the effects of resistance and plyometric exercises and high-intensity interval training (HIIT) on exercise economy [40,41]. Reduction in submaximal VO_2_ is significantly correlated with the reduced minute ventilation (VE) and heart rate (HR) [24].

The slope of VO_2_ (mL/min) to workload (Watts) demonstrates the economy of exercise and represents an indirect measure of cardiac output and aerobically generated ATP [23]. Commonly there is a continual rise throughout the exercise with the average slope of 10 mL/min/W with all exercise data [11]. Figure 1 depicts the VO_2_/workload responses of two athletes with different exercise economy. To evaluate an athlete’s exercise economy, it is better to measure it during an incremental test, which is fast enough to continually increase the VO_2_ similar to the intensities more commonly experienced during sports competition and at supra-LT workloads. In this regard, it is advisable to perform a CPET to measure the VO_2_ (mL/min) and to use a lower extremity ergometer to quantify the workloads (Watts) for proper measurement of VO_2_/workload slope.

### 3.4. Respiratory Compensation Point (RCP)

With increasing exercise intensity above the VT, the lactate production rate gets higher, and a point is reached when bicarbonate is no longer able to counteract exercise-induced metabolic acidosis. In the isocapnic buffering region, bicarbonate is decreasing with no evident hyperventilation. Then, there is an exponential increase in blood lactate concentration and an excess CO_2_, whereas the increase in VO_2_ remains linear. The second breakpoint in the ventilation response to exercise is where the peripheral chemoreceptors invoke hyperventilation, which is identified as the second VT or RCP [10,13,43,44].

According to the physiological changes at RCP, there is an inflection of VE versus VCO_2_ and also VE/VCO_2_ versus workload, so as depicted in Figure 2, the second VT is identifiable by the nadir of the VE/VCO_2_ to workload curve [43,44]. Both VE/VO_2_ and end-tidal O_2_ pressure (PETO_2_) increase while there is a deflection point on the PETCO_2_ trajectory [45]. It is usually achieved at around 70–80% of VO_2_max or 80–90% of peak HR during incremental exercise [13,43]. In a recent systematic review and meta-analysis, it has been demonstrated that there is a highly significant correlation between RCP and critical power, with the power output at CP being 6% lower than RCP [10,43,45].

The second VT or RCP can be expressed as VO_2_ (mL/kg/min) or percentage of VO_2_max. Workload consistent with RCP (Watts) might also provide a scale to compare the ability of different athletes to comply with higher intensities of exercise. Anaerobic capacity is an essential parameter in the performance of athletes, especially those who participate in sports with sudden bursts of high-intensity activity [4]. Therefore, a longer duration of exercise at a constant workload within the CP or at the RCP level (VO_2_ slow component) could be an advantage for such sportspersons.

Table 2 outlines the effects of training on each CPET variable in well-trained athletes compared to their ordinary counterparts.

## 4. Role of CPET in Diagnostic Workup

Many athletes experience some occasional vague symptoms such as exertional dyspnea, chest discomfort, and fatigue during their sports career in which the etiology could be cardiovascular, pulmonary, or muscular. Evaluating the physiological response of body organ systems to exercise provides valuable information on potential underlying ailments. In clinical practice, CPET is used to detect latent diseases and can help to differentiate cardiac and pulmonary problems. In elite endurance athletes, expiratory flow limitation (EFL) is assumed to be very frequent, with a prevalence of up to 40% in males and 90% in females [46]. The assessment of the flow-volume loop during CPET would reveal the presence and magnitude of EFL, further clarify a pulmonary mechanism for the symptoms and provide resolution of disease severity [16].

Assessment of ventilatory reserve and efficiency in addition to the standard haemodynamic and ECG monitoring provide insight into probable physiological abnormalities. The ventilatory reserve is the ratio between peak VE on a CPET and the maximum amount of air that can be breathed within one minute by a voluntary effort at rest termed the maximal voluntary ventilation (MVV), which is often measured in 15 s and multiplied by 4 [11,47]. Abnormal ventilatory reserve, which is VE/MVV ≥ 0.8 along with abnormalities in FEV1 and peak expiratory flow (PEF) are indicative of pulmonary limitations [11], but athletes with superior cardiovascular function can demonstrate some degrees of EFL and low ventilatory reserve with normal lung function tests [47]. Since EFL can be a cause of hypoxemia on exertion, pulse oximetry (SPO_2_) should also be measured throughout the CPET process. Ventilatory efficiency parameters [VE/VCO_2_ slope and PETCO_2_], reveal cardiopulmonary coupling and function, and when abnormal, may indicate subclinical ventilation-perfusion abnormalities as a possible mechanism for exertional symptoms [11,16]. Electrocardiographic and/or hemodynamic abnormalities like a hypertensive response to exercise or a slow recovery period might reveal a cardiovascular source for the symptoms.

Excessive training load without adequate recovery period exposes elite athletes to an inability to adjust optimally to the overall load. This process can results in overreaching, or in more severe cases overtraining syndrome (OTS), with different indeterminate signs and symptoms accompanying performance decrements and the development of acute illness [48]. Studies have shown that up to 64% of elite athletes experienced OTS at least once [49]. Parasympathetic alterations with bradycardia in endurance sports, and sympathetic alterations with tachycardia and hypertension in explosive and high intensity sports, have been suggested as various cardiovascular responses [50]. It is been advocated that heart rate and blood lactate concentration variations are the two most discriminating factors between overreached and normal athletes [51]. In an experimental study, Le Meur Y et.al. showed decreased cardiac output at submaximal and maximal exercise intensities with lower VO_2_max and reduced HR and SV values in triathletes after an overload training period [52]. It, therefore, seems that the CPET of elite athletes should be interpreted more carefully.

## 5. Exercise Prescription

CPET provides a context for determining a highly individualized training intensity zones for prescribing a structured exercise program. The physiological response to exercise characterizes the first and second VTs and VO_2_max, which allow for the identification of four intensity zones as it is illustrated in Figure 2 [43]. Heart rate and workload corresponding to each appropriate zone should be used for exercise prescription. Zone 1 consists of all workloads below the first VT, which represents light to moderate-intensity exercise. Zone 2 comprises those workloads between the first and second VT (RCP) equals to moderate to high-intensity exercise. The workloads above the CP that result in VO_2_max at exhaustion are in Zone 3 constitute high to sever intensity exercise domain. Sprints and all-out efforts above the workloads that allow for the attainment of VO_2_max are in Zone 4.

Constant workload exercise in Zone 1 brings about a steady-state VO_2_ that is sustainable for a long duration (>30 min) with only a modest sense of fatigue. It is suitable for the recovery phase of HIIT in athletes. Training in Zone 2 results in VO_2_ and lactate steady-state conditions and is important in inducing significant improvements in these parameters [24]. The highest workload with steady-state lactate is called critical power (CP), a marker of the upper-limit of sustainable prolonged aerobic exercise [43]. Endurance athletes like marathoners and triathletes benefit from improvements in these parameters.

Exercise training in Zones 3 and 4 might cause VO_2_ to reach maximum value without steady-state achievement and is better defined by CP concept (Time limit at VO_2_max). The duration of exercise is variable in these domains and based on the conditioning level of athletes, it would be in the range of 3 to 20 min in Zone 3 and less than that in Zone 4. Given a short exercise duration, these domains can only be used for HIIT programs [53]. There are various HIIT training protocols in the literature [53,54] that precisely characterize the physiological response of the exercise program.

## 6. Summary and Practical Implications

CPET has extensive practical applications in athletes, and comprehensive knowledge of exercise physiology in connection with various sports disciplines are essential for the interpretation of the results. Measurement of baseline fitness and the assessment of cardiopulmonary function in athletes suspected of having the cardiovascular or pulmonary disease are common indications for CPET. Revealing more talented athletes along with quantifying the physiologic parameters determinant of sports performance could be a reliable guide for team doctors and coaches.

VO_2_max and VT are the well-known fitness parameters that depend on several factors, including age, sex, genetic predisposition, and exercise training. Athletes need high values for best performance but with varying importance in different sports.

The critical power (CP) is a beneficial parameter for the assessment of an athlete’s endurance status. However, the ventilatory threshold (VT) is a suitable parameter in CPET to set the training loads in a highly individualized manner.O_2_ pulse indicates stroke volume and peripheral vascular perfusion/extraction response to exercise. The values at different levels of exercise demonstrate central and peripheral adaptations to exercise training.Ventilatory equivalent for oxygen (VE/VO_2_) indicates the ventilatory cost for O_2_ and begins to increase at the VT level. The values are lower at submaximal levels of exercise in well-trained athletes.Exercise economy is defined as ΔVO_2_/Δworkload, and the lower amounts are a marker of better endurance performance in elite athletes.Respiratory compensation point (RCP) somewhere called the second VT is when there is an exponential increase in VE in response to an excess CO_2_ production during incremental exercise. Critical power and, to a lesser extent RCP, represent the high to severe intensity of exercise and are useful in setting up an exercise training program for athletes in specific sport disciplines.

CPET is a well-accepted method to evaluate the function of body organs integrated into exercise. It could be a standard procedure to measure athletic performance in different sport disciplines but is disregarded in this field. Reference values have to be determined in athletes, so the interpretation of the results and performance differences would be accurately quantified. Hereby we call for further research on a large number of athletes in different disciplines to have comprehensive data for each CPET parameter in athletes.

## Figures and Tables

**Figure 1 jcm-10-05073-f001:**
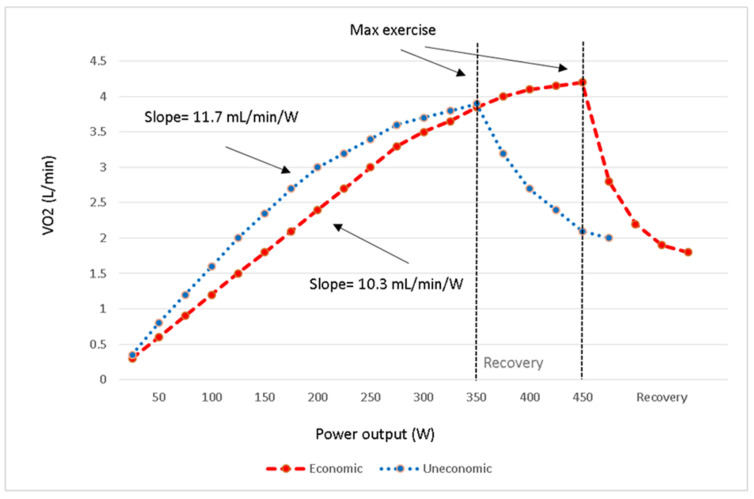
Athletes with different exercise economy as shown by VO_2_/workload responses to incremental exercise.

**Figure 2 jcm-10-05073-f002:**
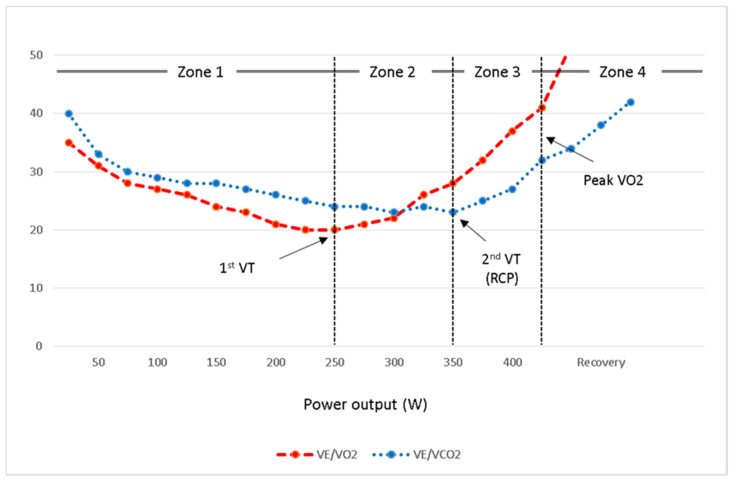
The ventilatory equivalents for oxygen (VE/VO_2_) and carbon dioxide (VE/VCO_2_) and their association with first and second VT which form four training zones during an incremental CPET. VT: ventilatory threshold, RCP: respiratory compensation point.

**Figure 3 jcm-10-05073-f003:**
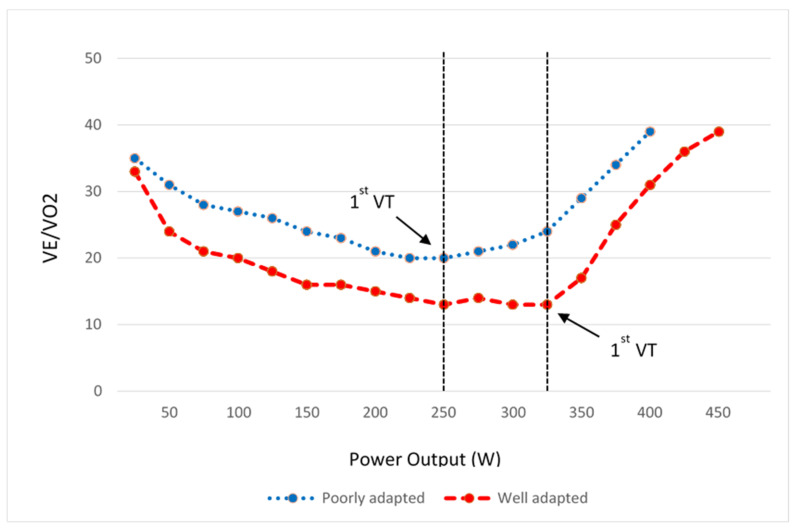
The ventilatory equivalent for O_2_ in a well-trained athlete compared with a poorly adapted athlete. As it’s evident, the ventilatory threshold is at higher workload with lower VE/VO_2_ values in well-trained athlete than their less fit peer.

**Table 1 jcm-10-05073-t001:** O_2_ pulse response of four athletes with different adaptation patterns to incremental exercise in comparison with a reference athlete

	Intensity of Exercise (% VO_2_max)		
	50%	75%	100%
Reference athlete O_2_ pulse (mL/beat)	14	18	20
Athlete A	↑	↑	↑
Athlete B	↑	↑↔	↓
Athlete C	↓	↓↔	↔↑
Athlete D	↓	↓	↓

Athlete A has a better central and peripheral adaptation to exercise than the reference athlete. Athlete B has better central but lower peripheral adaptation and Athlete C might has better peripheral adaptation than central. Athlete D is worst in both central and peripheral adaptations. Reference athlete: RA, ↑: more than RA, ↑↔: more or equals to RA, ↓↔: less or equals to RA, ↓: less than RA.

**Table 2 jcm-10-05073-t002:** Key CPET parameters in elite athletes.

	An Elite Athlete Compare to an Ordinary Peer
VO_2_ max (mL/kg/min)	↑↑
VO_2_ at VT	↑↑
Watts at VT	↑↑
O_2_ pulse/BSA	↑
VE/VO_2_ at VT	↓
VE/VCO_2_ slope	↔
PETCO_2_	↔
OUES	↔
ΔVO_2_/Δworkload	↔↓
VO_2_ at RCP	↑↑
Watts at RCP	↑↑
Exercise duration at RCP level	↑↑

RCP: respiratory compensation point, VT: ventilatory threshold, OUES: oxygen uptake efficiency slope, ↑↑: quite more, ↑: more, ↔: no difference, ↔↓: equals or less, ↓: less.

## Data Availability

All the data are available in the main text.

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
