# Peer review of "Cardiopulmonary Exercise Test Parameters in Athletic Population: A Review"

_jcm, 2021, doi:10.3390/jcm10215073_

Round 1

Reviewer 1 Report

In this narrative review CPET in athletes is described. It is a nice overview in which the major parameters of importance are described for athletes in comparison with healthy adults. The "new"information is limited, only other papers are reviewed.  

Major: I miss the description of the recovery phase (after max exercise) and how fast the recovery rate is for athletes compared to healthy adults. This could be added in most figures and tables.

Minors: 

line 13:  ..Although still underutilized.....: what is the evidence that this is underutilized. Are there hard data to prove that this is true?  Of course we could test all athletes in the world to find minor health deviations in an early stage but his would cost a huge amount of money with minor benefit: we could e.g. test better all "healthy" persons at high age (especially high risk patients) to measure their condition and detect diseases in this population.

 line 16: typo.. This manuscript demonstrates.....

line 23 That might limit sports participation? It is always useful to sport in a healthy way. Use the word  "overtraining"

line 30 remove the word ... correctly...

line 55 O2 pulse has 2 components (as correctly described further on in the manuscript) here only 1 component (stroke volume) is mentioned

line 78 VO2 max is the maximum oxygen uptake of the human body and thereby determines the maximal perfomance ability...

line 159: give evidence that division by BSA should be recommended (add a sentence and or give refs).

line178 (table 1) were are the values (14, 18, 20) from? (reference)

line 214 explain "remarkable economy" now the reader is forced to read ref 19

line 237 from which reference figure 3 is taken?

line 264  PETCO2 is trained athletes at max exercise is normally higher (they mentally keep on sporting so have higher lactate at max exercise and hyperventilate therefore more to compensate for the higher metabolic acidosis)

Author Response

Comments and Suggestions for Authors

Major: I miss the description of the recovery phase (after max exercise) and how fast the recovery rate is for athletes compared to healthy adults. This could be added in most figures and tables.

Response: The description of the recovery phase added to the VO2 max section of the paper and also added to the figures 1 & 3. Indeed except for VO2max, the normative CPET data for athletes in different disciplines is lacking, especially during the recovery phase. So it is difficult to provide data on athletes of various sports disciplines and compare them.

Minors:

Line 13:  ..Although still underutilized.....: what is the evidence that this is underutilized. Are there hard data to prove that this is true?  Of course we could test all athletes in the world to find minor health deviations in an early stage but his would cost a huge amount of money with minor benefit: we could e.g. test better all "healthy" persons at high age (especially high risk patients) to measure their condition and detect diseases in this population.

Response: In fact, except for VO2max and VT, there are just a few studies to address CPET parameters such as Oxygen pulse or exercise economy (ΔVO2/ΔWorkload) in athletes. As is mentioned in a recent paper by Prof. Aaron L Baggish*, CPET reference values for athletes have been lacking, and they developed a novel prediction equation for VO2peak of athletes. Other important parameters in CPET are really underutilized even in sports medicine clinics, and there are little data about them on professional athletes in the literature.

* https://pubmed.ncbi.nlm.nih.gov/34487164/

line 16: typo.. This manuscript demonstrates.....

Response: Corrected.

line 23: That might limit sports participation? It is always useful to sport in a healthy way. Use the word  "overtraining"

Response: The text corrected and overtraining syndrome is added to it. 

line 30: remove the word ... correctly...

Response: Removed.

line 55: O2 pulse has 2 components (as correctly described further on in the manuscript) here only 1 component (stroke volume) is mentioned

Response: The second component added to the text.

line 78: VO2 max is the maximum oxygen uptake of the human body and thereby determines the maximal perfomance ability...

Response: The definition of VO2max corrected.

line 159: give evidence that division by BSA should be recommended (add a sentence and or give refs).

Response: As in calculating the O2 pulse, VO2 should be in mL/min, the effect of athletes’ weight for making a reliable comparison is missing. Using the BSA for indexing the O2 pulse is just a suggestion from the authors of this paper. Description added to the text.

line178 (table 1) were are the values (14, 18, 20) from? (reference)

Response: The values in the tables and figures of the paper are just samples from the test results of athletes who came into the authors’ clinic.

line 214 explain "remarkable economy" now the reader is forced to read ref 19

Response: It added and corrected in the text.

line 237 from which reference figure 3 is taken?

Response: It is just samples from the authors’ clinic.

line 264  PETCO2 is trained athletes at max exercise is normally higher (they mentally keep on sporting so have higher lactate at max exercise and hyperventilate therefore more to compensate for the higher metabolic acidosis)

Response: As the data in table 2 are the parameters in elite athletes and actually compare athletes with their counterparts, the PETCO2 is almost the same between them. Indeed as is mentioned in the comment, PETCO2 in trained athletes at max exercise is different from the general non-athletic population.

Reviewer 2 Report

This review is well posited and perceived overall as valuable and the authors are to be congratulated.  Against this enthusiasm, however, are some points/perspectives that need to be improved to increase the relevance, clarity and potential impact of this work. 

Paramount among these are:

  1. In the Introduction some points raised regarding critical power/speed should be introduced. Later on where RCP is taken to represent the heavy-severe boundary the points regarding the agreement or error in estimating CP from RCP should be made.  See Galán-Rioja et al. (Sports Med. 2020 Oct;50(10):1771-1783).
  2. (~line 50) Please discuss the requirement to validate VO2max or not (Poole & Jones, J Appl Physiol (1985). 2017 Apr 1;122(4):997-1002).
  3. (~line 55). No, O2 pulse is not an important determinant of CO. HRxCV are.
  4. Very surprising that Karl Wasserman’s book (Principles of Exercise Testing and Interpretation. Lea & Febiger, Philadelphia, 1994. ) is not referenced as a standard in the Introduction.
  5. (line 78) No. VO2max does not define the maximal performance ability (CP+W’ does)

(lines 106-108) No! Please read recent Gladden/Rossiter/Brooks review (J Physiol. 2021 Feb;599(3):737-767) regarding lack of evidence for O2 supply limitation as determining the lactate threshold.

  1. (line 108) “exponentially” …more correctly would be “more steeply”.
  2. (line 133) But there is evidence that VT and LT can be dissociated (e.g., Hughes et al. J Appl Physiol Respir Environ Exerc Physiol 1982 Jun;52(6):1598-607.
  3. (line 154) “capillary conductivity”….more correct as “capillarity”
  4. (line 155) “higher a-vO2 during exercise”. See the D/betaQ analysis of Roca et al. (J Appl Physiol (1985). 1992 Sep;73(3):1067-76).
  5. (Line 162) Please clarify here exactly what is meant by “peripheral” versus “central”.
  6. (Line 182) “imperfect function” Please define meaning.
  7. (Line 184) “increased efficiency” is P:O ratio meant? Please define carefully.
  8. (Line 200). “weaker” perhaps should be “less fit”.
  9. (Line 228) “cardiovascular efficiency” means very different things to a cardiologist.
  10. (Line 235-) Redundant to line 217?
  11. (Line 232) Somewhere in here it should be specified that the rate of increase of WR on the incremental test should be fast enough to preclude development of VO2 slow component behavior and a steepening of the VO2 at supra-LT work rates.
  12. (Line 244) Please clarify. The isocapnic region is where HCO2- is decreasing – and also pH – but no hyperventilation (decreased PCO2) is evident.
  13. (Line 247) Please consider rewriting for clarity.
  14. (Line 254) “so-called critical power”. A little judgemental.
  15. (Line 255) See above as regards RCP relation to CP (Galán-Rioja et al. Sports Med. 2020 Oct;50(10):1771-1783).
  16. (Line 262) Please consider W’ from CP tests and the VO2 slow component in here.
  17. (Line 276-) Is expiratory flow limitation really a disease in highly fit athletes?
  18. (Line 282) MVV is often measured over 15s.
  19. (Line 300) “anaerobic sports” this convention belies that the energy used, even in very short events, is invariably replenished using molecular oxygen.
  20. (Line 315) Using the CP convention, all WRs >CP result in VO2max at exhaustion. The classic CP reviews all demonstrate such.
  21. (Line 326) But is defined by CP and W’! i.e. tlim = W’/(P-CP).
  22. (Line 342) Here and throughout. CP is probably a far better assessment of an athlete’s “endurance” status than VT.
  23. (Line 355) CP superior to MLSS and/or RCP (see Galán-Rioja et al. (Sports Med. 2020 Oct;50(10):1771-1783).

Minor

Line(s)

16          Why not start this sentence as : “CPET is valuable in clinical and physiological….”?

30          physicians and physiologists?

67          “peculiar conditions” might be better as “instances”

97          “44 to 72”

99          “fitness by about”

122        “VT by about”

244        “bicarbonate is”

262        “within the RCP”? If RCP is a discrete WR this doesn’t make sense.  Better to use CP?

316        When defined relative to CP it could be stated “constitute the heavy (<CP) to severe (>CP) intensity domains”? Perhaps usefully define “severe” exercise as that where, at exhaustion, VO2max is achieved.

Author Response

Comments and Suggestions for Authors

  1. In the Introduction some points raised regarding critical power/speed should be introduced. Later on where RCP is taken to represent the heavy-severe boundary the points regarding the agreement or error in estimating CP from RCP should be made. See Galán-Rioja et al. (Sports Med. 2020 Oct;50(10):1771-1783).

Response: Critical power added to the introduction section and also to the RCP section and referenced.

  1. (~line 50) Please discuss the requirement to validate VO2max or not (Poole & Jones, J Appl Physiol (1985). 2017 Apr 1;122(4):997-1002).

Response: Verification requirement of VO2max measurements added to the introduction section and referenced.

  1. (~line 55). No, O2 pulse is not an important determinant of CO. HRxCV are.

Response: Corrected.

  1. Very surprising that Karl Wasserman’s book (Principles of Exercise Testing and Interpretation. Lea & Febiger, Philadelphia, 1994. ) is not referenced as a standard in the Introduction.

Response: The Wasserman’s book added to the references.

  1. (line 78) No. VO2max does not define the maximal performance ability (CP+W’ does)

Response: Corrected.

(lines 106-108) No! Please read recent Gladden/Rossiter/Brooks review (J Physiol. 2021 Feb;599(3):737-767) regarding lack of evidence for O2 supply limitation as determining the lactate threshold.

Response: Definition of ventilatory threshold updated according to the new evidence and referenced.

  1. (line 108) “exponentially” …more correctly would be “more steeply”.

Response: Corrected.

  1. (line 133) But there is evidence that VT and LT can be dissociated (e.g., Hughes et al. J Appl Physiol Respir Environ Exerc Physiol 1982 Jun;52(6):1598-607.

Response: The possible discrepancy between VT and LT added to the text and referenced.

  1. (line 154) “capillary conductivity”….more correct as “capillarity”

Response: Corrected.

  1. (line 155) “higher a-vO2 during exercise”. See the D/betaQ analysis of Roca et al. (J Appl Physiol (1985). 1992 Sep;73(3):1067-76).

Response: It corrected and referenced.

  1. (Line 162) Please clarify here exactly what is meant by “peripheral” versus “central”.

Response: The description of peripheral and central added to the text.

  1. (Line 182) “imperfect function” Please define meaning.

Response: It corrected in the text.

  1. (Line 184) “increased efficiency” is P:O ratio meant? Please define carefully.

Response: It corrected in the text.

  1. (Line 200). “weaker” perhaps should be “less fit”.

Response: Corrected.

  1. (Line 228) “cardiovascular efficiency” means very different things to a cardiologist.

Response: It clarified based on the relevant reference.

https://pubmed.ncbi.nlm.nih.gov/26096801/

  1. (Line 235-) Redundant to line 217?

Response: Deleted.

  1. (Line 232) Somewhere in here it should be specified that the rate of increase of WR on the incremental test should be fast enough to preclude development of VO2 slow component behavior and a steepening of the VO2 at supra-LT work rates.

Response: It added to the text.

  1. (Line 244) Please clarify. The isocapnic region is where HCO2- is decreasing – and also pH – but no hyperventilation (decreased PCO2) is evident.

Response: It added to the text.

  1. (Line 247) Please consider rewriting for clarity.

Response: It corrected based on this reference. (Galán-Rioja et al. Sports Med. 2020 Oct;50(10):1771-1783)

  1. (Line 254) “so-called critical power”. A little judgemental.

Response: It corrected.

  1. (Line 255) See above as regards RCP relation to CP (Galán-Rioja et al. Sports Med. 2020 Oct;50(10):1771-1783).

Response: It corrected and referenced.

21: (Line 262) Please consider W’ from CP tests and the VO2 slow component in here.

Response: The VO2 slow component added instead of “Watt” to the text.

  1. (Line 276-) Is expiratory flow limitation really a disease in highly fit athletes?

Response: Indeed it is not a disease in athletes, but since the high profile athletes have improved cardiovascular function, some pulmonary limitations to exercise will be evident in those high levels of endurance exercise.

  1. (Line 282) MVV is often measured over 15s.

Response: It added to the text.

  1. (Line 300) “anaerobic sports” this convention belies that the energy used, even in very short events, is invariably replenished using molecular oxygen.

Response: It corrected based on the reference below.

https://pubmed.ncbi.nlm.nih.gov/27747843/

  1. (Line 315) Using the CP convention, all WRs >CP result in VO2max at exhaustion. The classic CP reviews all demonstrate such.

Response: It corrected just like the comment.

  1. (Line 326) But is defined by CP and W’! i.e. tlim = W’/(P-CP).

Response: It added to the text.

  1. (Line 342) Here and throughout. CP is probably a far better assessment of an athlete’s “endurance” status than VT.

Response: The CP added to the text and the VT corrected.

  1. (Line 355) CP superior to MLSS and/or RCP (see Galán-Rioja et al. (Sports Med. 2020 Oct;50(10):1771-1783).

Response: It corrected in the text.

Minor

Line(s)

16          Why not start this sentence as : “CPET is valuable in clinical and physiological….”?

Response: Corrected.

30          physicians and physiologists?

Response: It added to the text.

67          “peculiar conditions” might be better as “instances”

Response: Corrected.

97          “44 to 72”

Response: Corrected.

99          “fitness by about”

Response: Corrected.

122        “VT by about”

Response: Corrected.

244        “bicarbonate is”

Response: Corrected.

262        “within the RCP”? If RCP is a discrete WR this doesn’t make sense.  Better to use CP?

Response: Corrected.

316        When defined relative to CP it could be stated “constitute the heavy (<CP) to severe (>CP) intensity domains”? Perhaps usefully define “severe” exercise as that where, at exhaustion, VO2max is achieved.

Response: Corrected.